# Design and Analysis of Input Capacitor in DC–DC Boost Converter for Photovoltaic-Based Systems

Aamir Hayat [1], Daud Sibtain [2], Ali F. Murtaza [1], Sulman Shahzad [3], Muhammad Sheheryar Jajja [1] and Heybet Kilic [4,*]

1   Department of Electrical Engineering, University of Central Punjab, Lahore 54000, Pakistan
2   Energy Center of Excellence (ECoE), Harbin Electric International, Harbin 150028, China
3   Department of Electrical Engineering, Islamia University of Bahawalpur, Bahawalpur 63100, Pakistan; salmanshahzad05@gmail.com
4   Department of Electric Power and Energy Systems, Dicle University, 21280 Diyarbakır, Turkey
*   Correspondence: heybet.kilic@dicle.edu.tr

**Abstract:** Photovoltaic (P.V.) systems have become an emerging field for power generation by using renewable energy (RE) sources to overcome the usage of conventional combustible fuels and the massive release of dangerous gases. The efficient operation of the PV system is vital to extracting the maximum power from the PV source. For this, a maximum power point tracking (MPPT) algorithm works with a DC–DC converter to extract maximum power from the P.V. system. Two main issues may arise with the involvement of a converter: (1) to locate M.P.P and (2) the performance of the PV model in varying weather conditions. Therefore, designing any converter gain has the utmost significance; thus, the proposed work is on non-isolated boost converters. To calculate the values of specific parameters such as input capacitor, output capacitor, and inductor, the averaging state-space modeling typically uses governing equations. In this research, the formula of the input capacitor is derived through the average state-space modeling of the boost converter, which signifies the relation between input and output capacitors. From the results, it has been proven that the input capacitor efficiently performs when the input capacitor is half of the output capacitor. At an irradiance level of 1000 W/m$^2$, the system shows stable behavior with a fast convergence speed of 0.00745 s until the irradiance falls to a value of 400 W/m$^2$. The system is less stable during the morning and the evening when irradiance falls are very low.

**Keywords:** photovoltaic; maximum power point tracking; renewable energy; power electronics; input capacitor; output capacitor

## 1. Introduction

Energy demand is dramatically increasing with the increase in the population. The deficiency of fossil fuels has increased the world's concern about renewable sources [1]. In recent decades, people's main concern has been harvesting energy from photovoltaic systems [2]. However, the P.V. system shows nonlinear behavior during the day due to irradiance changes. Therefore, it is impossible to extract power from the photovoltaic (P.V.) system for 24 h [3]. Two basic strategies can achieve maximum power point tracking: establishing a hardware setup for the tracking of the sun and secondly by using software in association with a microcontroller to track the sun automatically in cooperation with power electronics [4].

The converters mainly used in M.P.P. systems are buck, boost, or buck-boost converters [5]. However, the converter is more frequently used in PV-based applications, and this proposed research is a boost converter due to its better stability and efficiency [6]. There are some other purposes for which maximum power point tracking (MPPT) uses the DC–DC converter. For example, it will match the load impedance to regulate the P.V. modules' input voltages and obtain maximum power transfer [7].

It is possible to reduce the steady-state error and transient response of the P.V. system through proper design and mathematical modeling of the converter. Furthermore, the P.V. system's settling time depends on the duty cycle and can be changed by changing the duty cycle of switching devices. The proposed boost converter design can enhance the available output voltage in the morning or evening when the fall of solar irradiance is low. Moreover, the boost converter contains the design of the inductor and capacitor and their related parameters. Because of the independence of the stand-alone system on the grid, the research of Cheng et al. was completed on a stand-alone system [8]. The characteristic curve of the PV panel is shown in Figure 1. The irradiance level varies from 1000 W/m$^2$ to 200 W/m$^2$ at 25 °C and Pmax.

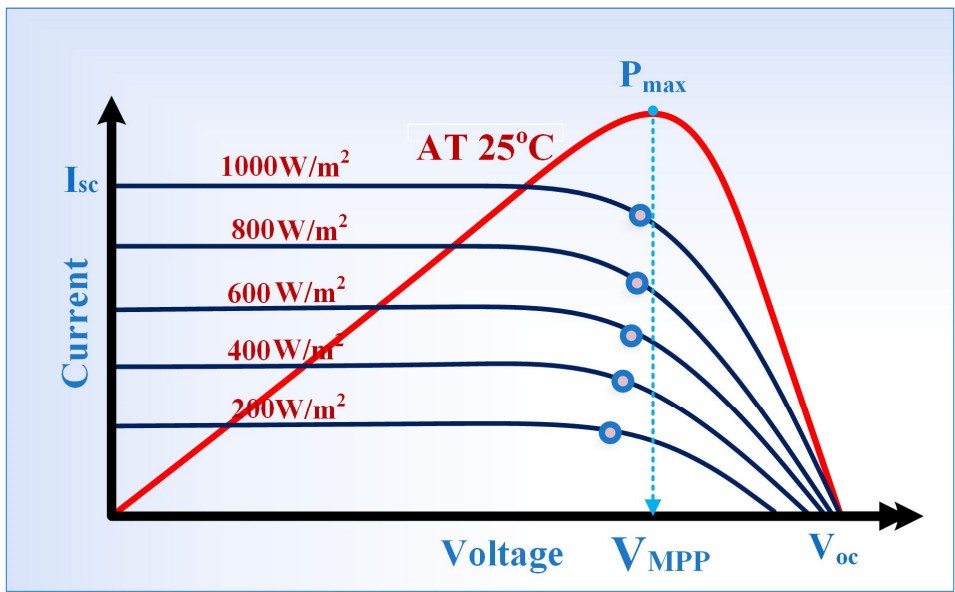

**Figure 1.** I-V and P-V characteristic curves.

DC–DC boost converters are widely used in photovoltaic (PV)-based systems because they can efficiently step up the input signal's voltage. The input capacitor is an essential component of the boost converter, as it stores energy and smooths out the output voltage. When the converter's switch is closed, the input capacitor is charged up, and when the switch is opened, the capacitor discharges its energy into the output. This helps to maintain a constant output voltage. Several studies have focused on designing and analyzing input capacitors in DC–DC boost converters for PV-based systems [9]. These studies have examined the effect of various capacitor parameters, such as capacitance, equivalent series resistance (ESR), equivalent series inductance (ESL), ripple current rating, and maximum voltage rating, on the converter's performance. A critical consideration in selecting an input capacitor for a boost converter is its capacitance per unit volume. This parameter determines how much energy the capacitor can store at a given physical size. Aluminum electrolytic capacitors have a higher capacitance per unit volume than other types, such as ceramic and tantalum capacitors. This makes them a good choice for PV-based systems, where space is often limited [10].

Another important consideration is the capacitor's ESR and ESL. ESR is a measure of the resistance of the capacitor's internal components, while ESL is a measure of the inductance of the capacitor's internal components. A capacitor with a low ESR and ESL will perform better in a boost converter, as it can more quickly and efficiently discharge its energy. Aluminum electrolytic capacitors are known for having relatively low ESR and ESL, making them a good choice for PV-based systems. The ripple current rating and maximum voltage rating are also important parameters to consider when selecting an input capacitor for a boost converter in a PV-based system. The ripple current rating determines the maximum current that the capacitor can handle without overheating [11,12].

On the other hand, the maximum voltage rating says how much voltage the capacitor can handle before it breaks. The capacitor will be able to handle more power and work more reliably if its ripple current rating and maximum voltage rating are both higher. Aluminum electrolytic capacitors are known for having high ripple current and maximum voltage ratings, making them a good choice for PV-based systems [13].

Much material has been discussed in the research world on the boost converter used in photovoltaic-based applications. Still, the discussion on the critical parameter, which is the design of the input capacitor, is limited [14]. Many researchers tried to determine the transfer function and use an input capacitor in the transfer function but failed to define the suitable value of the input capacitor. Keeping the constant current flow toward the load resistance while increasing the load resistance intensity gives rise to the available output voltage. This rise in voltage due to load variation results in a considerable increase in the output power produced. But practically, achieving the maximum available power is a big issue because the fall of solar radiation is finite in an exceptionally concentrated area. The permitted increase in load voltages is bound by certain limitations, and maximum output power is achieved by employing the maximum power point tracking (MPPT) strategy [15].

During this change in load, M.P.P. will occur at only one point. This condition will occur when the optimal load resistance (R.L) value equals the source resistance. It is not easy to choose an exact value of a static load that fulfills this condition. There is a need to minimize this problem so maximum power point trackers are used to obtain the maximum power point condition. If MPPT is fully or properly optimized, it can extract ninety-seven percent (97%) of photovoltaic power. The essential purpose of the maximum power point tracker is to equate or match the optimal impedance of the module to the load impedance [16,17]. The P.V. module's operating point depends on the impedance seen by the module, i.e., input impedance, when a DC–DC converter is placed between the load and the P.V. module. This input impedance depends on two primary parameters, duty cycle (d) and load resistance (R.L.). Now input resistance ($R_i$) can be easily changed by changing the duty cycle to match or equate optimal impedance at ambient conditions. There are a lot of methods for maximum power point tracker algorithm such as neural network, fuzzy logic control, ripple correlation control, a biological swarm chasing algorithm, hill climbing, conductance incremental, perturbation and observe (P&O), and look up Table [18,19]. To choose a suitable photovoltaic converter, some parameters should be considered. Sometimes, a buck is better than a boost, and vice versa, depending on their root mean square (R.M.S.) current through the input capacitor and output capacitor, specifications of MOSFET, diode, and inductor [20].

To compare the MOSFET of buck and boost, the switch's current rating is lower than buck's in the case of the boost converter. The problem of reverse current in the absence of light exists in the buck converter, so additional components (blocking diodes) are used to overcome this problem. Boost topology can be more suitable and efficient in the P.V. because it operates in continuous current mode, and it is more energy efficient than a buck converter. There are some limitations of boost converters used in P.V. A boost converter can only be operated if the load resistance ($R_L$) is less than or equal to the impedance of the module ($R_{MPP}$), and under low irradiations, a boost converter cannot track M.P.P. because it points in the non-operating region [21,22].

To consider the input capacitor, the buck converter needs an expansive and significant value of the capacitor to stabilize or smooth the discontinuity of the photovoltaic current. A small, low-cost capacitor can smooth the P.V. voltage and current because the boost converter current is as uniform as its inductor current. Similarly, in the case of an inductor, boost topology requires more inductance than buck topology [23]. The main contributions and novelties are highlighted below:

- Design of the input capacitor for the DC–DC boost converter by small-signal modeling of single-diode PV panel model, coupled to the boost converter via the input capacitor.
- Analysis of the converter under different irradiance levels by varying the capacitance of the input capacitor to extract the relationship between input and output capacitance.

- Stability analysis of the converter under different values of input and output capacitance.

The schematic diagram of the DC–DC boost converter is shown in Figure 2, where the PV source is connected to the load through a converter to meet the requirements of the load. Efficient converters will determine the performance of the PV panel.

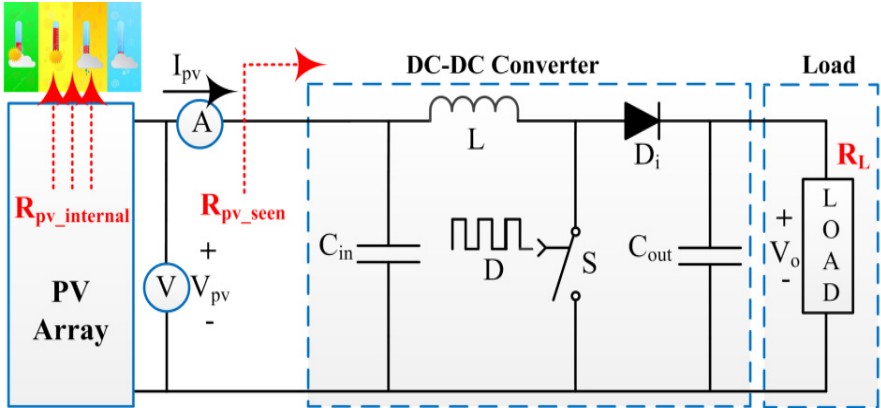

**Figure 2.** Schematic diagram of DC–DC boost converter with Ci.

The boost converter design components can be designed using formulas in Table 1, which represents the output capacitor and inductor design formulas at a ripple factor of 1%.

**Table 1.** Boost converter design parameters.

| Component | Ripple Factor | Components Formulas | Ref # |
|---|---|---|---|
| Output capacitor | 1% | $C_{out} = \dfrac{V_o D}{2\Delta V_{PV} R_L F_{SW}}$ | [2] |
| Inductor | 1% | $L = \dfrac{V_{in} D}{\Delta I_{IN} F_{SW}}$ | [2] |

## 2. Materials and Methods

The general equation of a state space model is given in Equations (1) and (2):

$$\dot{x} = Ax + Bu \tag{1}$$

$$y = Cx + Du \tag{2}$$

This system applies to linear and first-order differential equations, not nonlinear ones. Therefore, if there are some nonlinear components in the given system, then omit those nonlinear components and solve the remaining system. This will be executed by a switch (ON-OFF), called the system's small signal averaging. In the general equation of the state, space ($\dot{x}$) represents the first derivative, i.e., dx/dt. The boost converter will work in two switching modes such as:

i.  When the switch is ON.
ii. When the switch is OFF.

### 2.1. When the Switch Is ON

A power electronic device operates in ON and OFF modes. In ON mode, as shown in Figure 3, the current will flow through the switch, and the remaining circuit will be disconnected. At the input side, the P.V. array, which is a source of D.C. voltage ($V_{PV}$), and D.C. ($I_{PV}$) are attached. The current will pass through the inductor and capacitor. When this current flows through the inductor, the inductor will be energized, two voltages will appear on the output (source voltage and inductor voltage), and more voltages will appear on the output side. Similarly, when this current passes through the capacitor, the capacitor

will be charged up. Therefore, the mathematical modeling of the boost converter could be achieved by using state space modeling. In the next step, the averaging of these two state space equations and transfer function of the control parameter—which is duty cycle (d) and photovoltaic voltages ($V_{PV}$)—can be formulated. In this study, $i_L$ and $\dot{i_L}, \dot{V}_{PV}$ are used as state variables and $V_{PV}$ as an output variable. When the converter switch is in ON mode, the presentation of the DC–DC converter is shown in Figure 3.

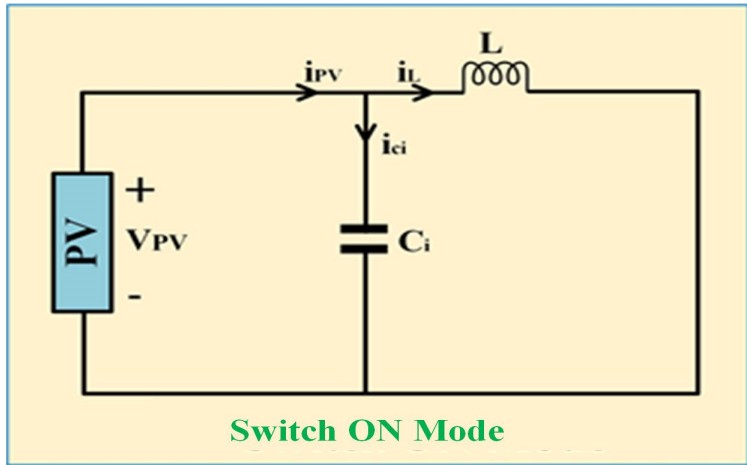

**Figure 3.** DC–DC converter in switch-on mode.

From Figure 3, the state space modeling is carried out as the inductor current equation is described in (3):

$$i_L = i_{pv} - i_{ci} \tag{3}$$

$$C_i = \frac{dv_{pv}}{dt} = -\frac{V_{pv}}{R_i} - i_L \tag{4}$$

$\because$ The -ve sign represents that the load is a source and $R_I$ is the source resistance.

$$\dot{v}_{pv} = -\frac{V_{pv}}{C_i R_i} - \frac{i_L}{C_i} \tag{5}$$

Then, we find the relationship of $\dot{i_L}$ the equation for the inductor by applying K.V.L.

$$L\frac{di_L}{dt} = V_{pv} \tag{6}$$

$$\dot{i_L} = \frac{1}{L}V_{pv} \tag{7}$$

The translation of Equations (5) and (7) into matrix form is described in (8).

$$\begin{bmatrix} \dot{i_L} \\ \dot{v}_{pv} \end{bmatrix} = \begin{bmatrix} 0 & \frac{1}{L} \\ -\frac{1}{C_i} & -\frac{1}{C_i R_i} \end{bmatrix} \begin{bmatrix} i_L \\ V_{pv} \end{bmatrix} + \begin{bmatrix} 0 \\ 0 \end{bmatrix} V_o \tag{8}$$

*2.2. When the Switch Is OFF*

When the switch is closed, the current will follow the other path, and at this time the diode conducts. One important point should be kept in mind that during OFF conditions, the path of the $I_{ci}$ reverse and, at this time, two currents ($I_{PV}$ and $I_{ci}$) will flow in the inductor. Once again, there are two further paths for inductor current ($I_L$), current ($I_{co}$) will flow through the output capacitor ($C_o$) and the remaining current ($I_o$) will flow through the load resistor ($R_L$) as shown in Figure 4.

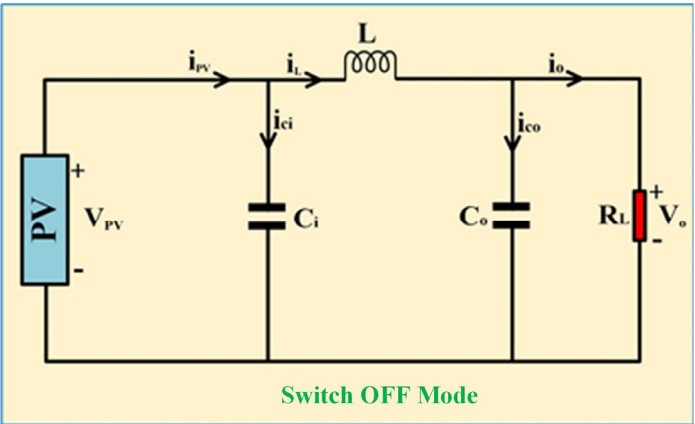

**Figure 4.** DC–DC converter in switch-off mode.

$$i_L = i_{pv} + i_{ci} \tag{9}$$

$$\frac{dV_{pv}}{dt} = -\frac{i_L}{C_i} - \frac{V_{pv}}{C_i R_i} \tag{10}$$

$$\dot{V}_{pv} = -\frac{i_L}{C_i} - \frac{V_{pv}}{C_i R_i} \tag{11}$$

The equation for the inductor is obtained by applying KVL in Figure 4.

$$L\frac{di_L}{dt} = V_{pv} - V_o \tag{12}$$

$$\dot{i}_L = \frac{V_{pv}}{L} - \frac{V_o}{L} \tag{13}$$

Similarly, Equations (11) and (13) are translated into matrix form as shown in (14).

$$\begin{bmatrix} \dot{i}_L \\ \dot{v}_{pv} \end{bmatrix} = \begin{bmatrix} 0 & \frac{1}{L} \\ -\frac{1}{C_i} & -\frac{1}{C_i R_i} \end{bmatrix} \begin{bmatrix} i_L \\ V_{pv} \end{bmatrix} + \begin{bmatrix} -\frac{1}{L} \\ 0 \end{bmatrix} V_o \tag{14}$$

$$\overline{A} = A_1 d + A_2(1-d) \tag{15}$$

$$\overline{A} = \begin{bmatrix} 0 & \frac{1}{L} \\ -\frac{1}{C_i} & -\frac{1}{C_i R_i} \end{bmatrix} d + \begin{bmatrix} 0 & \frac{1}{L} \\ -\frac{1}{C_i} & -\frac{1}{C_i R_i} \end{bmatrix} (1-d) \tag{16}$$

$$\overline{A} = \begin{bmatrix} 0 & \frac{1}{L} \\ -\frac{1}{C_i} & -\frac{1}{C_i R_i} \end{bmatrix} \tag{17}$$

Similarly, to calculate $\overline{B}$:

$$\overline{B} = B_1 d + B_2(1-d) \tag{18}$$

$$\overline{B} = \begin{bmatrix} 0 \\ 0 \end{bmatrix} d + \begin{bmatrix} -\frac{1}{L} \\ 0 \end{bmatrix} (1-d) \tag{19}$$

$$\begin{bmatrix} \dot{i}_L \\ \dot{v}_{PV} \end{bmatrix} = \begin{bmatrix} 0 & \frac{1}{L} \\ -\frac{1}{C_i} & -\frac{1}{C_i R_i} \end{bmatrix} \begin{bmatrix} i_L \\ V_{PV} \end{bmatrix} + \begin{bmatrix} -\frac{-(1-d)}{L} \\ 0 \end{bmatrix} V_o \tag{20}$$

The above-averaged matrices are written in the form of Equations (21) and (22).

$$\dot{i_L} = \frac{1}{L}V_{pv} - \frac{1-d}{L}V_o \tag{21}$$

$$\dot{v}_{pv} = -\frac{1}{C_i}i_L - \frac{1}{C_iR_i}V_{pv} \tag{22}$$

Introducing state variables:

$$i_L = I_L + \hat{i}_L, V_o = V_o + \hat{v}_o, d = D + \hat{d}, V_{pv} = V_{pv} + \hat{v}_{pv}$$

where $V_L$ is defined as

$$L\frac{di_l}{dt} = V_{pv} - (1-d)V_o$$

$$L\frac{d}{dt}(I_L + \hat{i}_L) = (V_{pv} - \hat{v}_{pv}) - \left(1 - (D - \hat{d})\right)(V_o + \hat{v}_{pv})$$

Equate AC and DC quantities and proceed with the AC equation (neglect second-order AC quantities).

$$L\frac{d}{dt}(\hat{i}_L) = \hat{v}_{pv} + \hat{d}V_o - D\hat{v}_o \tag{23}$$

Suppose $(1-D)\hat{v}_o = 0$.

$$L\frac{d}{dt}(\hat{i}_L) = \hat{v}_{pv} + \hat{d}V_o \tag{24}$$

Taking the Laplace transform of Equation (24):

$$\hat{i}_L(s) = \frac{\hat{v}_{pv}(s)}{sL} + \frac{V_o\hat{d}(s)}{sL} \tag{25}$$

Using Equation (22):

$$\dot{v}_{pv} = -\frac{i_L}{C_i} - \frac{V_{pv}}{C_iR_i} \tag{26}$$

$$C_i\frac{d}{dt}(V_{PV} + \dot{v}_{pv}) = -(I_L - \hat{i}_L) - \frac{1}{R_i}(V_{PV} + \hat{v}_{pv}) \tag{27}$$

$$C_i\frac{d(\hat{v}_{pv})}{dt} = -\hat{i}_L - \frac{\hat{v}_{pv}}{R_i} \tag{28}$$

Taking Laplace on both sides of Equation (28):

$$C_i\frac{d(\hat{v}_{pv})}{dt} = -\hat{i}_L - \frac{\hat{v}_{pv}}{R_i} \tag{29}$$

Substituting the value of $\hat{i}_L(s)$ from Equation (25) in Equation (29):

$$sC_i\hat{v}_{PV}(s) = -\hat{i}_L(s) - \frac{\hat{v}_{pv}(s)}{R_i} \tag{30}$$

$$\frac{\hat{v}_{pv}(s)}{d(s)} = -\frac{V_o}{(LC_i)s^2 + \frac{sL}{R_i} + 1} \tag{31}$$

Using the relation $R_i = (1-D)^2 R_L$ in the above expression and finally, obtain the transfer function to control (d cycle) to photovoltaic voltage ($V_{PV}$).

$$G_{VP,d} = \frac{-V_o}{(LC_i)s^2 + \left(\frac{L}{R_L(1-d)^2}\right)s + 1} \tag{32}$$

Natural frequency ($w_n$) and damping ratio ($\zeta$) are two main parameters to determine the stability of any system. These two parameters can be calculated by comparing the transfer function with the general formulas of these two parameters and comparing these values with the given transfer function, which is a second-order transfer function.

### 2.3. Design of Input Capacitor

We compare the coefficients of the transfer function given in (32) with the general form of the second-order transfer function in (33) and the damping factor.

$$G_s = \frac{1}{ms^2 + bs + k} \tag{33}$$

$$\omega_n = \sqrt{\frac{k}{m}} \tag{34}$$

$$\omega_n = \sqrt{\frac{1}{LC_i}} \tag{35}$$

Therefore, the natural frequency value can be calculated using Equation (35). Where L is the inductor and $C_i$ is the input capacitor. The damping factor can be calculated from Equation (36).

$$\zeta = \frac{1}{2}\left[\frac{b}{\sqrt{\frac{k}{m}}}\right] \tag{36}$$

Compare Equations (29) and (35) to obtain the expression of the input capacitor.

$$\zeta = \frac{1}{2}\left(\frac{L}{R_L(1-d)^2}\right)\frac{1}{\sqrt{LC_i}} \tag{37}$$

$$\zeta^2 = \frac{1}{4R_{MPP}{}^2} \times \frac{L}{C_i} \tag{38}$$

Rearrange Equation (38) to find $C_i$.

$$C_i = \frac{1}{4(R_{MPP})^2} \times \frac{L}{\zeta^2} \tag{39}$$

From Equation (39), it is concluded that the input capacitor is mainly dependent upon the inductor (L), damping factor ($\zeta$), and $R_{MPP}$. Moreover, it is a generalized formula that can be applied to calculate the value of the input capacitor at any irradiance.

## 3. Results and Analysis

The proposed design of the input capacitor is analyzed under different scenarios discussed in forthcoming sections.

### 3.1. Input Capacitor Equal to Output Capacitor ($C_i$ = 45 μF, $C_o$ = 45 μF, L = 12 mH)

To further evaluate the performance of the PV system by equating the value of the input capacitor ($C_i$) to output capacitor ($C_o$). The system shows stable behavior during the irradiances ranging from 1000 to 500 W/m$^2$ with fast convergence speed of 0.00767 s for

1000 W/m$^2$. Furthermore, it shows oscillation below 500 W/m$^2$ with a slow convergence speed of 0.0156 s, as shown in Figure 5. Figure 5 shows the system's response varying from 1000 W/m$^2$ to 200 W/m$^2$, and the efficiency reduces to zero with further variation in weather conditions. It can be seen from Figure 5 that the damping response of the system is more dominant at low irradiance levels as compared with that of high irradiance levels, and at low irradiance levels below 400 W/m$^2$, the settling time of the system is high as compared with that of high irradiance levels.

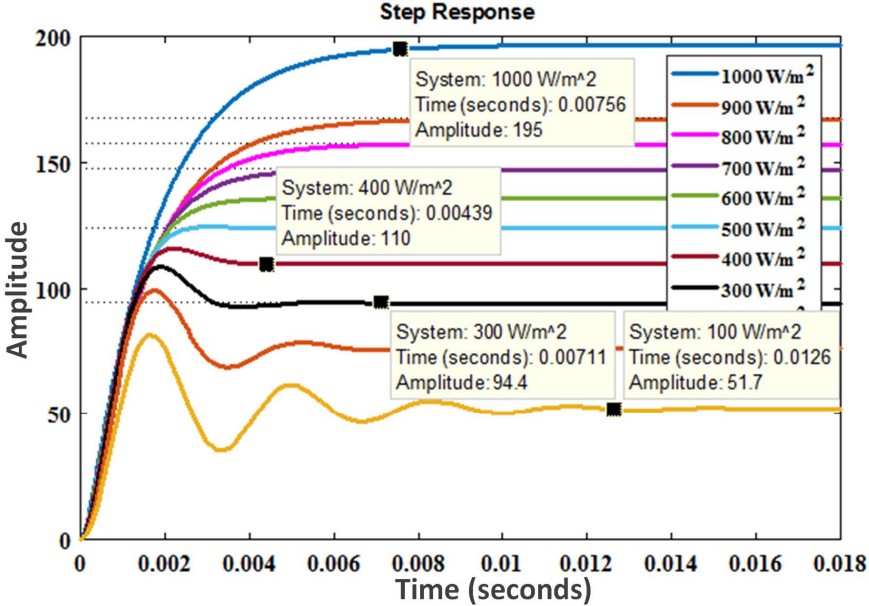

**Figure 5.** Step response at $C_i$ equal to $C_o$.

The P-Z map of the model mentioned above has been validated in Figure 6. At the start, the fall of irradiations is 1000 W/m$^2$ at peak hours; the system shows stable behavior until the irradiation level decreases from 1000 to 500 W/m$^2$. During this range of irradiances, the pole lies on the real axis. Any further reduction in irradiance, such as 400 to 100 W/m$^2$, conjugate poles are formed as revealed in Table 2. At 100 to 200 W/m$^2$ (during morning and evening time), the system is briefly stable and can shift the conjugate poles in the right half plane.

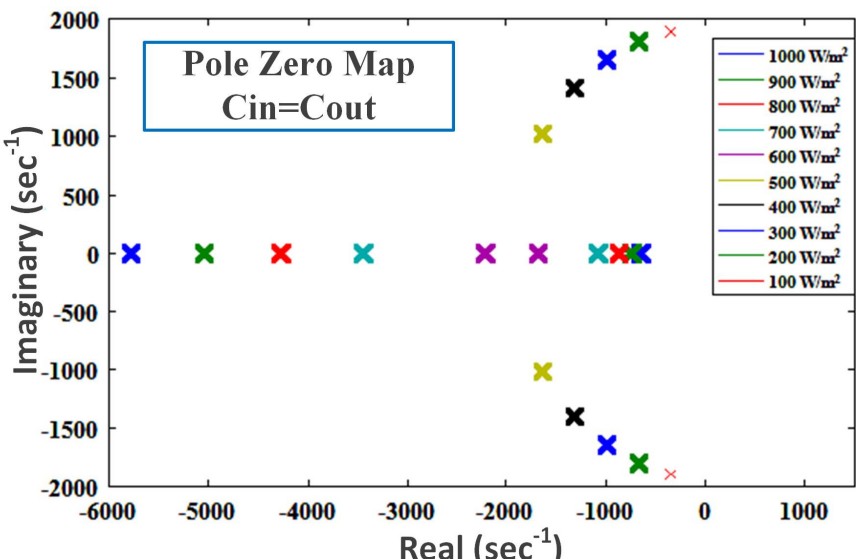

**Figure 6.** P-Z Map at $C_i$ equal to $C_o$.

**Table 2.** $C_i$ equal to $C_o$.

| Irradiance Level (W/m$^2$) at 25 °C | Damping Factor | Transient Time (s) | Poles | Zeros |
|---|---|---|---|---|
| 1000 | 3.0005 | $2.6851 \times 10^{-4}$ | $1.0 \times 10^4, -1.3271 - 0.0279$ | NIL |
| 900 | 3.0110 | $2.8885 \times 10^{-4}$ | $1.0 \times 10^4, -1.1228 - 0.0330$ | NIL |
| 800 | 2.6807 | $3.1196 \times 10^{-4}$ | $1.0 \times 10^3, -9.9030 - 0.3740$ | NIL |
| 700 | 2.3510 | $3.5572 \times 10^{-4}$ | $1.0 \times 10^3, -8.6126 - 0.4300$ | NIL |
| 600 | 2.0215 | $4.1371 \times 10^{-4}$ | $1.0 \times 10^3, -7.2586 - 0.5103$ | NIL |
| 500 | 1.6924 | $4.9415 \times 10^{-4}$ | $1.0 \times 10^3, -5.8939 - 0.6284$ | NIL |
| 400 | 1.3626 | $6.1373 \times 10^{-4}$ | $1.0 \times 10^3, -4.4075 - 0.8403$ | NIL |
| 300 | 1.0317 | $8.1056 \times 10^{-4}$ | $1.0 \times 10^3, -2.4628 - 1.5039$ | NIL |
| 200 | 0.6972 | 0.0012 | $1.0 \times 10^3, -1.3404 + 1.3810i, -1.3404 - 1.3810i$ | NIL |
| 100 | 0.3559 | 0.0024 | $1.0 \times 10^3, -0.6851 + 1.7984i, -0.6851 - 1.7984i$ | NIL |

The step response summary and the quantitative analysis of the P-Z map were performed. The location of the pole and damping factor against the varying irradiance levels are analyzed in Table 2.

*3.2. Input Capacitor Half of the Output Capacitor ($C_i$ = 22.5 μF, $C_o$ = 45 μF, L = 12 mH)*

Now consider another case when $C_i$ is taken as half of the value of $C_o$ at 22.5 μF and 45 μF, respectively. During peak hours, the irradiance level is 1000 W/m$^2$, and the system shows stable behavior with a fast convergence speed of 0.00745 s until the irradiance falls to the value of 400 W/m$^2$, as shown in Figure 7. The system is less stable during the morning and evening when irradiance falls are very low.

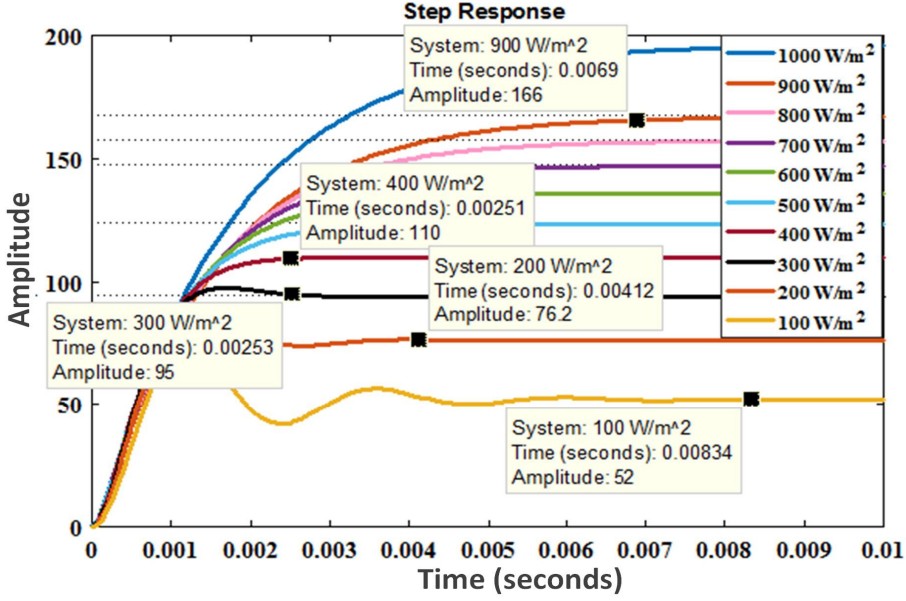

**Figure 7.** Step response when $C_i$ is less than $C_o$.

Similarly, the P-Z map is plotted for the model's validity, as shown in Figure 8. It can be analyzed from Figure 8 that the stability of the system is further increased. The load resistance was set at 15 Ωι, and the switching frequency of the boost converter was set at 20 KHz. To configure the inductor and output capacitor of the boost converter, a 1% ripple factor [2] was selected. The two designs to configure L and $C_o$ boost converters [2] were

selected from the literature shown in Table 1. However, as far as $C_i$ is concerned, none has provided any formula to configure $C_i$. Therefore, each of these designs was provided with $C_i$, whose value is configured according to the proposed criteria. Each design of the boost converter has been discussed with two scenarios: (1) with the proposed $C_i$ value and (2) half of the proposed $C_i$ value. Then, the performance of different designs is evaluated (Table 3).

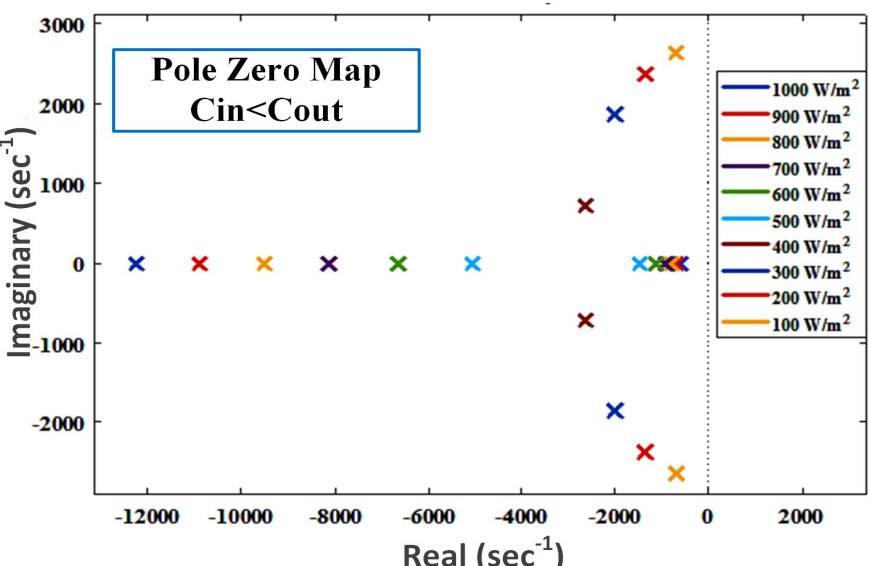

**Figure 8.** P-Z plot at $C_i$ half of $C_o$.

**Table 3.** $C_i$ half of $C_o$.

| Irradiance Level (W/m$^2$) at 25 °C | Damping Factor | Transient Time (s) | Poles | Zeros |
|---|---|---|---|---|
| 1000 | 2.1217 | $5.5743 \times 10^{-4}$ | $1.0 \times 10^3, -6.4899 - 0.2853$ | NIL |
| 900 | 2.1291 | $5.5549 \times 10^{-4}$ | $1.0 \times 10^3, -5.4386 - 0.3405$ | NIL |
| 800 | 1.8956 | $6.2393 \times 10^{-4}$ | $1.0 \times 10^3, -4.7485 - 0.3900$ | NIL |
| 700 | 1.6624 | $7.1144 \times 10^{-4}$ | $1.0 \times 10^3, -4.0659 - 0.4555$ | NIL |
| 600 | 1.4294 | $8.2741 \times 10^{-4}$ | $1.0 \times 10^3, -3.3280 - 0.5565$ | NIL |
| 500 | 1.1967 | $9.8830 \times 10^{-4}$ | $1.0 \times 10^3, -2.5289 - 0.7323$ | NIL |
| 400 | 0.9635 | 0.0012 | $1.0 \times 10^3, -1.3120 + 0.3614i, -1.3120 - 0.3614i$ | NIL |
| 300 | 0.7295 | 0.0016 | $1.0 \times 10^2, -9.9166 + 9.3191i, -9.9166 - 9.3191i$ | NIL |
| 200 | 0.4930 | 0.0024 | $1.0 \times 10^3, -0.6702 + 1.1844i, -0.6702 - 1.1844i$ | NIL |
| 100 | 0.2516 | 0.0047 | $1.0 \times 10^3, -0.3425 + 1.3170i, -0.3425 - 1.3170i$ | NIL |

Further analysis was carried out when $C_i$ is double the output capacitor. The step response is plotted from irradiance of 1000 to 100 W/m$^2$, and convergence analysis was carried out. It is clear from the results, as shown in Figure 9, the system has achieved some instability. Furthermore, the irradiance below 600 W/m$^2$ started showing more oscillation than other conditions analyzed before. Response at 100 W/m$^2$ shows the worst response, and its settling time is also increased to around 0.0305 s.

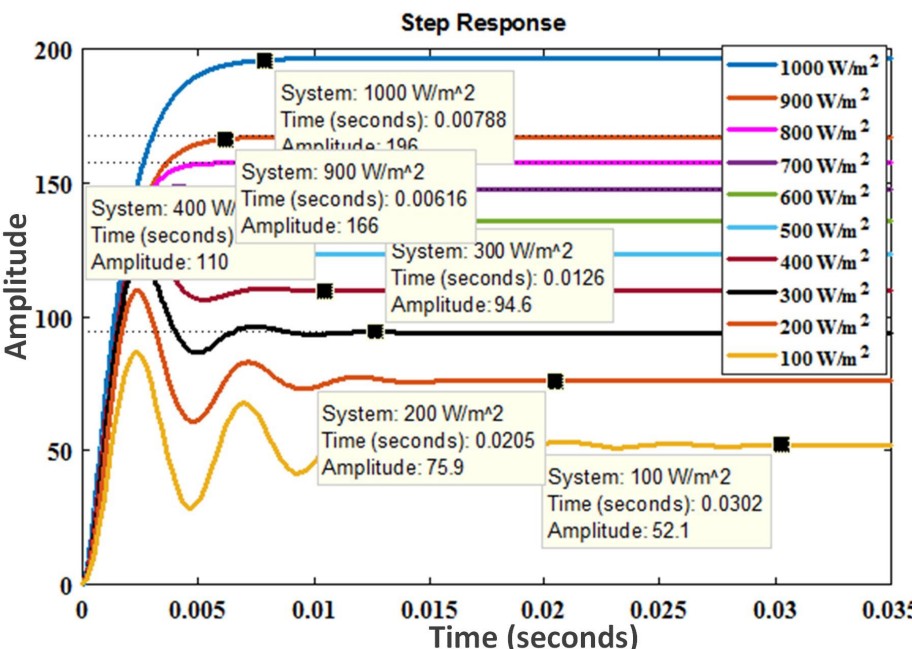

**Figure 9.** Step response at $C_i$ is greater than $C_o$.

To check the validity of this scenario, the P-Z map is plotted as shown in Figure 10. It can be analyzed from the graph that several poles are shifted towards the imaginary axis. It adds oscillations in the system, and it can be seen that the poles from 100 to 600 W/m² are far away from the imaginary axis, which exhibits more fluctuations than the other irradiance conditions.

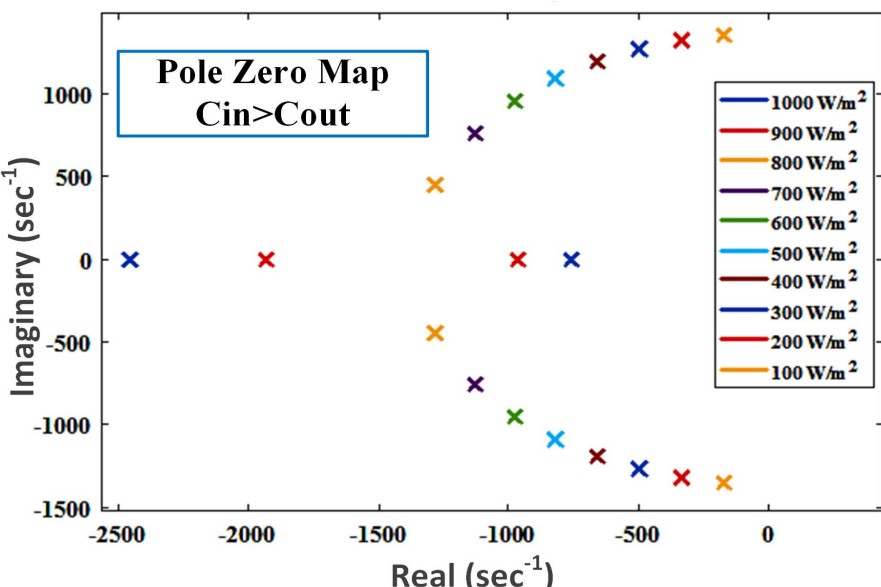

**Figure 10.** P-Z plot for $C_i$ greater than $C_o$.

In addition, the combined analysis of the above conditions is illustrated under decreasing irradiance conditions, as shown in Figure 11. The close-up section at points A and B is analyzed, and it can be visualized that when the input capacitance is less than the output capacitance, the decaying curve follows the utmost ideal case. The other two states diverge more from their ideal states, indicating ultimate power loss during tracking.

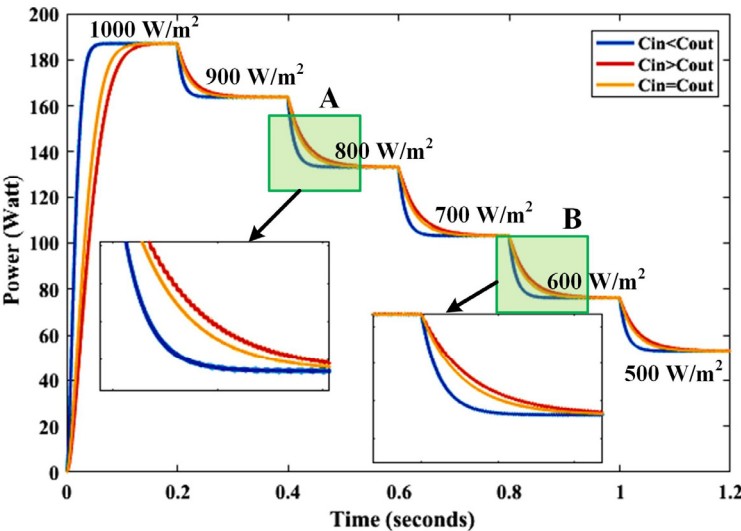

**Figure 11.** Test under decreasing irradiance.

Similarly, a further test was carried out for varying irradiance levels, as shown in Figure 12. Analysis at increasing irradiance was monitored, and it can seem that when the input capacitance is less than the output ($C_i < C_o$), tracking of MPP is the nearest to the ideal state.

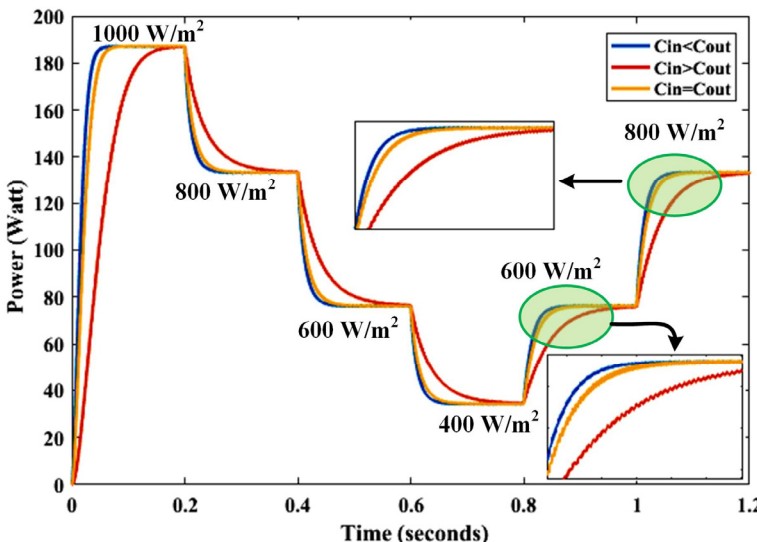

**Figure 12.** Test under varying irradiance.

Furthermore, the impact of load variation and switching frequency is analyzed in Figures 13 and 14. As load varies from the rating the converter is designed power extraction from the source varies. Similarly, the switching frequency has an impact on the size of the components, which can be analyzed from the equations as discussed earlier.

Gain margin, over-damped output, and transient response are all enhanced when the input capacitor is less in value relative to the output capacitor. However, the suggested DC–DC boost converter system's performance and behavior are negatively impacted when the input capacitor value is greater than the output capacitor value, which in turn impacts the quality of the output voltage and current.

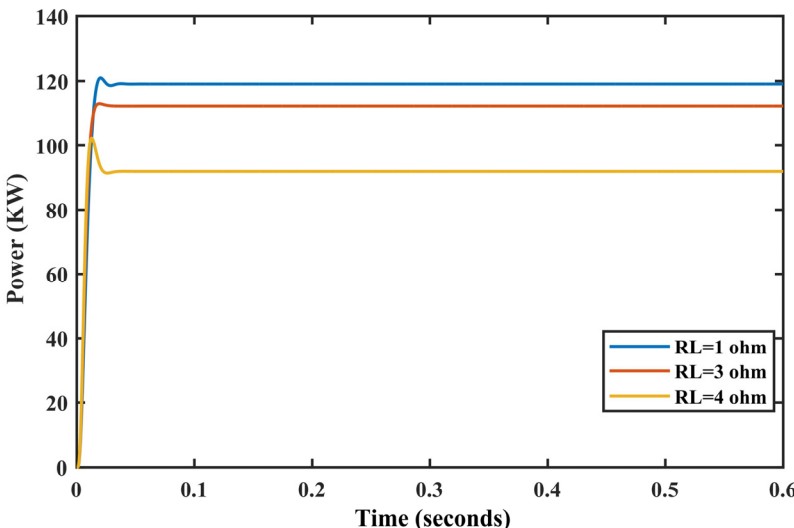

**Figure 13.** PV response against load variations.

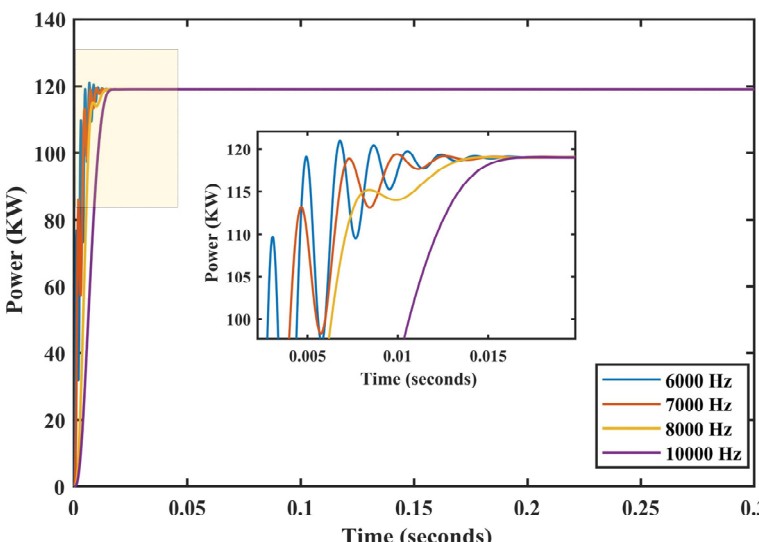

**Figure 14.** PV system response against switching frequency variations.

## 4. Conclusions

The design and analysis of the input capacitor in boost converters for PV-based systems have been extensively studied in the literature. The input capacitor value significantly impacts the converter's performance, and various methods have been proposed for selecting the optimal input capacitor value. Moreover, the input capacitor has been analyzed through mathematical modeling, which can be used to predict the converter's stability and dynamic response. These findings can help design and optimize boost converters for PV-based systems. Moreover, the boost converter for photovoltaic arrays gives better results when the input capacitor is used at different values of $C_i$ to validate the efficiency of the system response under ambient conditions. The results proved that the critical factor in achieving desired results is the selection of $C_i$, at which overall efficiency is mainly dependent. The lower value of $C_i$ results in better efficiency and transient response. On the other hand, the high value of $C_i$ relative to the value of $C_o$ results in unsatisfactory system performance. Finally, it is concluded that the stability of the boost converter is purely dependent on the value of $C_i$, and from the simulation results, it can be determined that the $C_i$ should be 45 µF at 300 W/m$^2$ for the proposed system. The designed formula for $C_i$ is a generalized formula, and the value of $C_i$ can be changed according to solar irradiance

worldwide according to irradiance level. The value of $C_i$ should be equal to or less than $C_o$ to achieve better results.

In future studies, the input capacitor can easily be selected by exploring this research work. In addition, different control algorithms can be evaluated by varying the capacitance value to improve the performance of the stand-alone PV system. This research work can be further extended by analyzing the performance of different MPPT algorithms by changing the values of the capacitance under different weather conditions.

**Author Contributions:** Conceptualization, A.H. and D.S.; methodology, A.F.M.; software, S.S.; writing—original draft preparation, M.S.J.; writing—review and editing, D.S. and A.H.; supervision & project administration, H.K. All authors have read and agreed to the published version of the manuscript.

**Funding:** This work received funding from the Department of Electric Power and Energy System, Dicle University, 21280, Diyarbakır, Turkey.

**Institutional Review Board Statement:** Not applicable.

**Informed Consent Statement:** Not applicable.

**Data Availability Statement:** Not applicable.

**Conflicts of Interest:** The authors declare no conflict of interest.

## Abbreviations

| | |
|---|---|
| $R_L$ | Load resistance ($\Omega_l$) |
| $w_n$ | Natural frequency (Hz) |
| $\zeta$ | Damping ratio |
| $C_i$ | Input capacitor (µF) |
| $C_o$ | Output capacitor (µF) |
| L | Inductance (mH) |
| MPPT | Maximum power point tracking |
| P.V. | Photovoltaic |
| ESR | Equivalent series resistance |
| ESL | Equivalent series inductance |
| P-Z | Pole zero |
| RE | Renewable energy |

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
