# Peer review of "Design and Analysis of Input Capacitor in DC–DC Boost Converter for Photovoltaic-Based Systems"

_sustainability, doi:10.3390/su15076321_

Round 1

Reviewer 1 Report

The paper is interesting but needs many corrections:

1.. Abstract must be improved to consider the problem, objective, methodology, findings, and main contribution and results.

2.. The bibliographic references must be improved and updated, several recent papers from 2017 to 2023 must be cited. In his paper it is observed that more than half of the cited papers are very old.

3.. Avoid capitalizing the first letter of all keywords.

4.. In all the tables you are not using the format defined in the journal for the tables.

5.. Use the same font for the symbols, for example in line 94 use bold font for RL, but later in the formulas it does not use that same font, make this correction throughout the document.

6.. The figures 3, 4, 12, 11, 10, 8, 7, 6, 5 and in general all the figures must improve and must be unified, for example they must have the size of the letters, avoid strong colors, use the same type letter you use in the equations. Avoid using large fonts.

7.. In line 220 use another font for Cl, Co, L, you must use the same font you use in the equations.

8.. In figure 12 you must make a space between the number and the units, for example, make this correction throughout the document.

9.. Results must be added for different load values.

10.. An analysis and results must be added for different switching frequencies of the switch.

11.. The conclusions should be improved, it can be added based on the contributions and objectives of the paper.

12.. A list of contributions of the paper should be made, please add it at the end of the introduction.

13.. It is recommended to use a better equation editor

Reviewer 2 Report

Equation (39) should be one of key contributions in this work. However, the variable 'd' was not well explained.

Will the results in this work be different if the capacity of the panel, i.e. the charging current, varies?

All '2' to represent the square of area were not imposed.

More quantitative results should be included, especially the conclusions section.

Reviewer 3 Report

1- Avoid using abbreviations in the keywords.

2- The main subject o the paper has not been discussed in the literature review part.

3- The English is poor and needs a native English editor to improve it.

Equation (39) is a generalize formula  --- Did the authors mean general or generalized?

So many other words and expressions should be rewritten.

4- The contribution of the paper is not clear.

5- To highlight the contribution, the review part should emphasize the knowledge gap between published works and the present one.

6- In figures 7 and 9, precise the y-axis nomenclature (amplitude). 

7- I it is from another source, the authors should cite the reference of the figure 1.

8- The quality of the figures should be improved.

9- What is the origin of the instability shown in figures 7 and 9 for low irradiances?

The choice of input capacitance is critical to the proper operation of the converter but as it is presented, the contribution and results of the paper are not sufficient.

Round 2

Reviewer 1 Report

Thank you very much for the corrections

Author Response

Thank you for your response

Reviewer 2 Report

Ln 254, the zeta should be delta. Instead, the damping ratio should be zeta.

Ln 352, the second word 'it' should be deleted.

Author Response

Comment-1: Ln 254, the zeta should be delta. Instead, the damping ratio should be zeta.

Response: Thank you for pointing out this issue which is actually arises during writing all equation in the previous revision. There is no concept of delta in this paper, it was zeta actually (ζ) which reflects the damping factor which was replaced with other symbol like delta, now the equations from 36-39 are corrected and the statement is also rectified.

Comment-2: Ln 352, the second word 'it' should be deleted.

Response: Thank you for highlighting the error which has been corrected now.

Reviewer 3 Report

The manuscript has been improved. The following comments should be addressed.

1- In various conditions, there can be low irradiances. As the authors said: "if the irradiance level fall below 400 W/m2 the system show underdamped response, and it takes more time to achieve stability. This effect is more prominent when value of input capacitor is greater than the output capacitor".

How can the authors guarantee that this issue is not a source of the whole PV system performance decrees?

2- From line 175 to line 251, the method of formulating the linking sentence between equations is very poor. Some sentences are without verbs and/or meaningless.

3- For the whole manuscript, extensive editing of the English language and style is extremely required.

4- Before the conclusions section, the authors should add the main recommendations based on the main findings of the work according to the studied cases.

5- The references should be written according to the journal requirements.

Author Response

The manuscript has been improved. The following comments should be addressed:

Comment-1: In various conditions, there can be low irradiances. As the authors said: "if the irradiance level fall below 400 W/m2 the system show underdamped response, and it takes more time to achieve stability. This effect is more prominent when value of input capacitor is greater than the output capacitor. How can the authors guarantee that this issue is not a source of the whole PV system performance decrees?

Response: As per the understanding of the comment the impact of the input capacitor is more prominent at low irradiance level and this impact directly reflects the performance of the PV system. Moreover, the overall performance of the PV will be improved by using the input capacitor of rating less than the output capacitor to achieve better performance in term of response time with minimum power loss, as also showcased in Figure-11 and Figure-12 (in the manuscript The analysis is all about the impact of the input capacitor and the performance of the PV-based dc-dc converter. When the value of Cin<Cout or Cin=Cout the under damped response is minimized as compared to Cin> Cout. Hence, apply the relation Cin<Cout improves the performance of the PV system. 

Comment-2: From line 175 to line 251, the method of formulating the linking sentence between equations is very poor. Some sentences are without verbs and/or meaningless.

Response:  Thank you for highlighting the issue. The comment has been addressed.

Comment-3: For the whole manuscript, extensive editing of the English language and style is extremely required.

Response: In light of the comment, the whole manuscript has been revised to improve the English issues.

Comment-4: Before the conclusions section, the authors should add the main recommendations based on the main findings of the work according to the studied cases.

Response: Thank you for the comment. As per the suggestion, the main findings are summarized for a better understanding of the result section.

Comment-5: The references should be written according to the journal requirements.

Response: Thank you for the comment. The references have been revised in accordance with journal requirements.